

Microbial decomposition processes and vulnerable Arctic soil organic carbon in the 21st century
Junrong Zha and Qianlai Zhuang
Department of Earth, Atmospheric, and Planetary Sciences and Department of Agronomy,
Purdue University, West Lafayette, IN 47907 USA
Correspondence to: qzhuang@purdue.edu



**Abstract**
**Inadequate representation of biogeochemical processes in current biogeochemistry models**
**contributes to a large uncertainty in carbon budget quantification. Here, detailed microbial**
**mechanisms were incorporated into a process-based biogeochemistry model, the Terrestrial**
**Ecosystem Model (TEM). Ensemble regional simulations with the model estimated the**
**carbon budget of the Arctic ecosystems is 76.0±114.8 Pg C during the 20th century, -**
**3.1±61.7 Pg C under the RCP 2.6 scenario and 94.7±46 Pg C under the RCP 8.5 scenario**
**during the 21st century.  Positive values indicate the regional carbon sink while negative**
**values are source to the atmosphere.  Compared to the estimates using a simpler soil**
**decomposition algorithm in TEM, the new model estimated that the Arctic terrestrial**
**ecosystems stored 12 Pg less carbon over the 20th century, 19 Pg C and 30 Pg C less under**
**the RCP 8.5 and RCP 2.6 scenarios, respectively, during the 21st century.  When soil carbon**
**within depths 30 cm, 100 cm and 300 cm was considered as initial carbon in the 21st**
**century simulations, the region was estimated to accumulate 65.4, 88.6, and 109.8 Pg C,**
**respectively, under the RCP 8.5 scenario. In contrast, under the RCP 2.6 scenario, the**
**region lost 0.7, 2.2, and 3 Pg C, respectively, to the atmosphere.  We conclude that the**
**future regional carbon budget evaluation largely depends on whether or not the adequate**
**microbial activities are represented in earth system models and the sizes of soil carbon**
**considered in model simulations.**





## 1. Introduction

Northern high-latitude soils and permafrost contain more than 1,600 Pg carbon (Tarnocai et al 2009). Climate over this region has warmed in recent decades (Serreze and Francis 2006) and the increase is 1.5 to 4.5 times the global mean (Holland and Bitz 2003). Warming-induced changes in carbon cycling are expected to exert large feedbacks to the global climate system (Davidson and Janssens 2006, Christensen and Christensen 2007, Oechel et al 2000).

Warming is expected to accelerate soil C loss by increasing soil respiration, but increasing nutrient mineralization, thereby stimulating plant net primary production (NPP) (Mack et al 2004, Hobbie et al 2002). Thus, the variation of climate may switch the role of the Arctic system between a C sink and a source if soil C loss overtakes NPP (Davidson et al 2000, Jobbágy and Jackson 2000). Process-based biogeochemical models such as TEM (Hayes et al 2014, Raich and Schlesinger 1992, McGuire et al 1992, Zhuang et al 2001, 2002, 2003, 2010, 2013), Biome-BGC (Running and Coughlan 1988), CASA (Potter et al 1993), CENTURY (Parton et al 1994) and Biosphere Energy Transfer Hydrology scheme (BETHY) (Knorr et al 2000) have been widely used to quantify the response of carbon dynamics to climatic changes (Todd-Brown et al 2012). An ensemble of process-based model simulations suggests that arctic ecosystems acted as a sink of atmospheric $CO_2$ in recent decades (McGuire et al 2012, Schimel et al 2013). However, the response of this sink to increasing levels of atmospheric $CO_2$ and climate change is still uncertain (Todd-Brown et al 2013). The IPCC 5[th] report also shows that land carbon storage is the largest source of uncertainty in the global carbon budget quantification (Ciais et al 2013).



Much of the uncertainty is also due to the inadequate representation of ecosystem
processes that determine the exchanges of water, energy and C between land ecosystems and the
atmosphere (Wieder et al 2013), and ignorance of some key biogeochemical mechanisms
(Schmidt et al 2011). For example, heterotrophic respiration ($R_H$) is the primary loss pathway for
soil organic carbon (Hanson et al 2000, Bond-Lamberty and Thomson 2010). and it generally
increases with increasing temperature (Davidson and Janssens 2006) and moisture levels in well-
drained soils (Cook and Orchard 2008). Moreover, this process is closely related to soil nitrogen
mineralization that determines soil N availability and affects gross primary production (Hao et al
2015). To date, most models treated soil decomposition as a first-order decay process, i.e., $CO_2$
respiration is directly proportional to soil organic carbon. However, it is not clear if these models
are robust under changing environmental conditions (Lawrence et al 2011, Schimel and
Weintraub 2003, Barichivich et al 2013) since they often ignored the effects of changes in
biomass and composition of decomposers, while recent empirical studies have shown that
microbial abundance and community play a significant role in soil carbon decomposition
(Allison and Martiny 2008). The control that microbial activity and enzymatic kinetics imposed
on soil respiration suggests the need for explicit representation of microbial physiology,
enzymatic activity, in addition to the direct effects of soil temperature and soil moisture on
heterotrophic respiration (Schimel and Weintraub 2003). Recent mechanistically-based models
explicitly incorporated with the microbial dynamics and enzyme kinetics that catalyze soil C
decomposition have produced notably different results and a closer match to contemporary
observations (Wieder et al 2013, Allison et al 2010) indicating the need for incorporating these





microbial mechanisms into large-scale earth system models to quantify carbon dynamics under
future climatic conditions ((Wieder et al 2013, Allison et al 2010).

This study advanced a microbe-based biogeochemistry model (MIC-TEM) based on an

extant Terrestrial Ecosystem Model (TEM) (Raich and Schlesinger 1992, McGuire et al 1992,
Zhuang et al 2001, 2002, 2003, 2010, 2013, Hao et al 2015). In MIC-TEM, the heterotrophic
respiration is not only a function of soil temperature, soil organic matter (SOM) and soil
moisture, but also considers the effects of dynamics of microbial biomass and enzyme kinetics
(Allison et al 2010). The verified MIC-TEM was used to quantify the regional carbon dynamics
in northern high latitudes (north 45 °N) during the 20$^{th}$ and 21$^{st}$ centuries.

**2.   Methods**
**2.1 Overview**
Below we first briefly describe how we advanced the MIC-TEM by modifying the soil
respiration process in TEM (Zhuang et al 2003) to better represent carbon dynamics in terrestrial
ecosystems. Second, we describe how we parameterized and verified the new model using
observed net ecosystem exchange data at representative sites and how simulated net primary
productivity (NPP) was evaluated with Moderate Resolution Imaging Spectroradiometer
(MODIS) data to demonstrate the reliability of new model at regional scales. Third, we present
how we applied the model to the northern high latitudes for the 20$^{th}$ and 21$^{st}$ centuries. Finally,
we introduce how we conducted the sensitivity analysis on initial soil carbon input, using
gridded observation-based soil carbon data of three soil depths during the 21$^{st}$ century.



## 2.2 Model description

TEM is a highly aggregated large-scale biogeochemical model that estimates the dynamics of

carbon and nitrogen fluxes and pool sizes of plants and soils using spatially referenced

information on climate, elevation, soils and vegetation (Raich and Schlesinger 1992, McGuire et

al 1992, Zhuang et al 2003, 2010, Melillo et al 1993). To explicitly consider the effects of

microbial dynamics and enzyme kinetics on large-scale carbon dynamics of northern terrestrial

ecosystems, we developed MIC-TEM by coupling version 5.0 of TEM (Zhuang et al 2003, 2010)

with a microbial-enzyme module (Hao et al 2015, Allison et al 2010). Our modification of the

TEM improved the representation of the heterotrophic respiration ($R_H$) from a first-order

structure to a more detailed structure (Fig. S1).

In TEM, heterotrophic respiration $R_H$ is calculated as a function of soil organic carbon

(SOC), soil temperature ($Q_{10}$), soil moisture (f (MOIST)), and the gram-specific decomposition

constant $K_d$:

$$R_H = K_d * SOC * Q_{10}^{\frac{DT}{10}} * f(MOIST) \qquad (1)$$

where DT is soil temperature at top 20 cm. $CO_2$ production from SOC pool is directly

proportional to the pool size, and the activity of decomposers only depends on the built-in

relationships with soil temperature and moisture (Todd-Brown et al 2012). Therefore, the

changes in microbial community composition or adaption of microbial physiology to new

conditions were not represented in TEM. However, current studies indicate that soil C

decomposition depends on the activity of biological communities dominated by microbes

(Schimel and Weintraub 2003), implying that the biomass and composition of the decomposer

community can't be ignored (Todd-Brown et al 2012).



We thus revised the first-order soil C structure in TEM to a second-order structure
considering microbial dynamics and enzyme kinetics according to Allison et al 2010. In MIC-
TEM, heterotrophic respiration ($R_H$) is calculated as:

$$R_H = ASSIM*(1-CUE)$$                     (2)

Where ASSIM and CUE represent microbial assimilation and carbon use efficiency, respectively.
ASSIM is modeled with a Michaelis-Menten function:

$$ASSIM = Vmax_{uptake} * MIC * \frac{DOC}{Km_{uptake}+DOC}$$         (3)

Where $Vmax_{uptake}$ is the maximum velocity of the reaction and calculated using the Arrhenius
equation:

$$Vmax_{uptake} = Vmax_{uptake_0} * e^{-\frac{Ea_{uptake}}{R*(temp+273)}}$$         (4)

$Vmax_{uptake_0}$ is the pre-exponential coefficient, $Ea_{uptake}$ is the activation energy for the reaction
($Jmol^{-1}$), R is the gas constant (8.314 $Jmol^{-1}K^{-1}$), and temp is the temperature in Celsius under the
reaction occurs.
Besides, $Km_{uptake}$ value is calculated as a linear function of temperature:

$$Km_{uptake} = Km_{uptake_{slope}} * temp + Km_{uptake_0}$$         (5)

Microbial biomass MIC is modeled as:

$$\frac{dMIC}{dt} = ASSIM * CUE - DEATH - EPROD$$         (6)

Where microbial biomass death (DEATH) and enzyme production (EPROD) are modeled as
constant fraction of microbial biomass:

$$DEATH = r_{death} * MIC$$         (7)

$$EPROD = r_{EnzProd} * MIC$$         (8)





Where $r_{death}$ and $r_{EnzProd}$ are the ratio of microbial death and enzyme production, respectively.

Dissolved organic carbon (DOC) is part of soil organic carbon:

$$\frac{dDOC}{dt} = DEATH * (1 - MICtoSOC) + DECAY + ELOSS - ASSIM \quad (9)$$

where MICtoSOC is carbon input as dead microbial biomass to SOC, representing the fraction of
microbial death that flows into SOC, and is set as a constant value according to Allison et al
2010. SOC dynamics are modeled:
$$\frac{dSOC}{dt} = Litterfall + DEATH * MICtoSOC - DECAY \quad (10)$$

Where Litterfall is estimated as a function of vegetation carbon (Zhuang et al 2010). The
enzymatic decay of SOC is calculated as:
$$DECAY = V_{max} * ENZ * \frac{SOC}{Km+SOC} \quad (11)$$

Where $V_{max}$ is the maximum velocity of the reaction and calculated using the Arrhenius equation:
$$V_{max} = Vmax_0 * e^{-\frac{Ea}{R*(temp+273)}} \quad (12)$$

The parameters Km and carbon use efficiency (CUE) are temperature sensitive, and calculated
as a linear function of temperature between 0 and 50 ℃:
$$Km = Km_{slope} * temp + Km_0 \quad (13)$$

$$CUE = CUE_{slope} * temp + CUE_0 \quad (14)$$

Where CUEslope and $CUE_0$ are parameters for calculating CUE. The values of $CUE_{slope}$ and
$CUE_0$ were derived from Allison et al 2010.
ELOSS is also a first-order process, representing the loss of enzyme:
$$ELOSS = r_{enzloss} * ENZ \quad (15)$$

Where $r_{enzloss}$ is the ratio of enzyme loss. Enzyme pool (ENZ) is modeled:



$$\frac{dENZ}{dt} = EPROD\text{-}ELOSS \qquad (16)$$

Heterotrophic respiration ($R_H$) is an indispensable component of soil respiration (Bond-
Lamberty and Thomson 2010), and closely coupled with soil nitrogen (N) mineralization that
determines soil N availability, affecting gross primary production (GPP).


**2.3 Model parameterization and validation**
The variables and parameters of these microbial dynamics and their impacts on soil C
decomposition were detailed in Allison et al 2010 (Table S5). Here we parameterized MIC-TEM
for representative ecosystem types in northern high latitudes based on monthly net ecosystem
productivity (NEP, $gCm^{-2}$ $mon^{-1}$) measurements from AmeriFlux network (Davidson et al 2000)
(Table S1). The results for model parameterization was presented in Fig. S2. Another set of level
4 gap-filled NEP data was used for model validation at site level (Table S2). The site-level
monthly climate data of air temperature ( ℃), precipitation (mm) and cloudiness (%) were used
to drive the model. Gridded MODIS NPP data from 2001 to 2010 were used to evaluate regional
NPP simulation.
The parameterization was conducted with a global optimization algorithm SCE-UA
(Shuffled complex evolution) (Duan et al 1994) to minimize the difference between the monthly
simulated and measured NEE at these sites (Fig. S2). The cost function of the minimization is:
$$Obj = \sum_{i=1}^{k}(NEP_{obs,i} - NEP_{sim,i})^2 \qquad (17)$$

Where $NEP_{obs,i}$ and $NEP_{sim,i}$ are the observed and simulated NEP, respectively. k is the number
of data pairs for comparison. Other parameters used in MIC-TEM were default values from TEM



5.0 (Zhuang et al 2003, 2010). The optimized parameters were used for model validation and
regional extrapolations.

**2.4 Regional simulations**
Two sets of regional simulations for the 20th century using MIC-TEM and TEM at a spatial
resolution of $0.5\,°$ latitude $\times 0.5\,°$ longitude were conducted. Gridded forcing data of monthly air
temperature, precipitation, and cloudiness were used, along with other ancillary inputs including
historical atmospheric $CO_2$ concentrations, soil texture, elevation, and potential natural
vegetation. Climatic inputs vary over time and space, whereas soil texture, elevation, and land
cover data are assumed to remain unchanged throughout the 20th century, which only vary
spatially. The transient climate data during the 20th century was organized from the Climatic
Research Unit (CRU TS3.1) from the University of East Anglia (Harris et al 2014). The spatial-
explicit data include potential natural vegetation (Melillo et al 1993), soil texture (Zhuang et al
2003) and elevation (Zhuang et al 2015).

Similarly, two sets of simulations were conducted driven with two contrasting climate

change scenarios (RCP 2.6 and RCP 8.5) over the 21st century. The future climate change
scenarios were derived from the HadGEM2-ES model, which is a member of CMIP5 project
(https://esgf-node.llnl.gov/search/cmip5/). The future atmospheric $CO_2$ concentrations and
climate forcing from each of the two climate change scenarios were used. The simulated NPP, $R_H$
and NEP by both models (TEM 5.0 and MIC-TEM) were analyzed. The positive NEP represents
a $CO_2$ sink from the atmosphere to terrestrial ecosystems, while a negative value represents a
source of $CO_2$ from terrestrial ecosystems to the atmosphere.





Besides, in order to test the parameter uncertainty in our model, we conducted the
regional simulations with 50 sets of parameters for both historical and future studies. The 50 sets
of parameters were obtained according to the method in Tang and Zhuang 2008. The upper and
lower bounds of the regional estimations were generated based on these simulations.

**2.5 Sensitivity to initial soil carbon input**
Future carbon dynamics can be affected by varying initial soil carbon amount. In the standard
simulation of TEM, the initial soil carbon amount for transient simulations was obtained from
equilibrium and spin-up periods directly for each grid cell in the region. To test the sensitivity to
the initial soil carbon amount in transient simulations for the 21$^{st}$ century, we used empirical soil
organic carbon data extracted from the Northern Circumpolar Soil Carbon Database (NCSCD)
(Tarnocai et al 2009), as the initial soil carbon amount. The $0.5\,° \times 0.5\,°$ soil carbon data products
for three different depths of 30cm, 100cm and 300cm were used. The sensitivity test was
conducted for transient simulations under the RCP 2.6 and RCP 8.5 scenarios. To avoid the
instability of C-N ratio caused by replacing the initial soil carbon pool with observed data at the
beginning of transient period, initial soil nitrogen values were also generated based on the soil
carbon data and corresponding C-N ratio map for transient simulations (Zhuang et al 2003, Raich
and Schlesinger 1992).

**3.  Results**
**3.1 Model verification at site and regional levels**



With the optimized parameters, MIC-TEM reproduces the carbon dynamics well for alpine
tundra, boreal forest, temperate coniferous forest, temperate deciduous forest, grasslands and wet
tundra with $R^2$ ranging from 0.70 for Ivotuk to 0.94 for Bartlett Experimental Forest (Fig. S3,
table S3). In general, model performs better for forest ecosystems than for tundra ecosystems.
The temporal NPP from 2001 to 2010 simulated by MIC-TEM and TEM were compared with
MODIS NPP data (Fig. S4). Pearson correlation coefficients are 0.52 (MIC-TEM and MODIS)
and 0.34 (TEM and MODIS). NPP simulated by MIC-TEM showed higher spatial correlation
coefficients with MODIS data than TEM (Fig. S5). By considering more detailed microbial
activities, the heterotrophic respiration is more adequately simulated using the MIC-TEM. The
simulated differences in soil decomposition result in different levels of soil available nitrogen,
which influences the nitrogen uptake by plants, the rate of photosynthesis and NPP. The spatial
correlation coefficient between NPP simulated by MIC-TEM and MODIS is close to 1 in most
study areas, suggesting the reliability of MIC-TEM at the regional scale.

**3.2 Regional carbon dynamics during the 20th century**
The equifinality of the parameters in MIC-TEM was considered in our ensemble regional
simulations to measure the parameter uncertainty (Tang and Zhuang 2008). Here and below, the
ensemble means and the inter-simulation standard deviations are shown for uncertainty measure,
unless specified as others. These ensemble simulations indicated that the northern high latitudes
act from a carbon source of 38.9 PgC to a carbon sink of 190.8 PgC by different ensemble
members, with the mean of 64.2±21.4 Pg at the end of 20th century while the simulation with the
optimized parameters estimates a regional carbon sink of 77.6 Pg with the interannual standard



deviation of 0.21 PgC yr$^{-1}$ during the 20$^{th}$ century (Fig 1). Simulated regional NEP with
optimized parameters using TEM and MIC-TEM showed an increasing trend throughout the 20$^{th}$
century except a slight decrease during the 1960s (Fig. 2). The Spatial distributions of NEP
simulated by MIC-TEM for different periods in 20$^{th}$ century also show the increasing trend (Fig
3). Positive values of NEP represent sinks of $CO_2$ into terrestrial ecosystems, while negative
values represent sources of $CO_2$ to the atmosphere. From 1900 onwards, both models estimated a
regional carbon sink during the 20$^{th}$ century. With optimized parameters, TEM estimated higher
NPP and R$_H$ at 0.6 PgC yr$^{-1}$ and 0.3 PgC yr$^{-1}$ than MIC-TEM, respectively, at the end of the 20$^{th}$
century (Fig. 2). The MIC-TEM estimated a carbon sink increase from 0.64 to 0.83 PgCyr$^{-1}$
during the century while the estimated increase by TEM was much higher (0.28 PgCyr$^{-1}$) (Fig. 2).
At the end of the century, MIC-TEM estimated NEP reached 1.0 PgCyr$^{-1}$ in comparison with
TEM estimates of 0.3 PgCyr$^{-1}$. TEM estimated NPP and R$_H$ are 0.5 PgCyr$^{-1}$ and 0.3 PgCyr$^{-1}$
higher, respectively. As a result, TEM estimated that the region accumulated 11.4 Pg more
carbon than MIC-TEM. Boreal forests are a major carbon sink at 0.55 and 0.63 PgCyr$^{-1}$
estimated by MIC-TEM and TEM, respectively. Alpine tundra contributes the least sink. Overall,
TEM overestimated the sink by 12.5% in comparison to MIC-TEM for forest ecosystems and
16.7% for grasslands. For wet tundra and alpine tundra, TEM overestimated about 20% and 33%
in comparison with MIC-TEM, respectively (Table 1).

**3.3 Regional carbon dynamics during the 21$^{st}$ century**
Regional annual NPP and R$_H$ increases under the RCP 8.5 scenario according to simulations with
both models (Fig. 4). With optimized parameters, MIC-TEM estimated NPP increases from 9.2



in the 2000s to 13.2 PgCyr$^{-1}$ in the 2090s, while TEM predicted NPP is 2.0 PgCyr$^{-1}$ higher in the
2000s and 0.3 PgCyr$^{-1}$ higher in the 2090s (Fig. 4). Similarly, TEM also overestimated $R_H$ by 1.7
PgCyr$^{-1}$ in the 2000s and 0.25 PgCyr$^{-1}$ higher in the 2090s, respectively (Fig. 4). As a result, the
regional sink increases from 0.53 PgCyr$^{-1}$ in the 2000s, 1.4 PgCyr$^{-1}$ in the 2070s, then decreases
to 1.1 PgCyr$^{-1}$ in the 2090s estimated by MIC-TEM (Fig. 4). Given the uncertainty in parameters,
MIC-TEM predicted the region acts as a carbon sink ranging from 48.7 to 140.7 Pg, with the
mean of 71.7±26.6 Pg at the end of 21$^{st}$ century, while the simulation with optimized parameters
estimates a regional carbon source of 79.5 Pg with the interannual standard deviation of 0.37
PgC yr$^{-1}$ during the 21$^{st}$ century (Fig 4). TEM predicted a similar trend for NEP, which
overestimated the carbon sink with magnitude of 19.2 Pg compared with the simulation by MIC-
TEM with optimized parameters. Under the RCP 2.6 scenario (Fig. 4), the increase of NPP and
$R_H$ is smaller from 2000 to 2100 compared to the simulation under the RCP 8.5. MIC-TEM
predicted that NPP increases from 9.1 to 10.9 PgCyr$^{-1}$, TEM estimated 1.6 PgCyr$^{-1}$ higher at the
beginning and 0.9 PgCyr$^{-1}$ higher in the end of the 21$^{st}$ century (Fig. 4). Consequently, MIC-
TEM predicted NEP fluctuates between sinks and sources during the century, with a neutral
before 2070, and a source between -0.2 - -0.3 Pg C yr$^{-1}$ after the 2070s. As a result, the region
acts as a carbon source of 1.6 Pg C with the interannual standard deviation of 0.24 PgC yr$^{-1}$
estimated with MIC-TEM and a sink of 27.6 Pg C with the interannual standard deviation of 0.2
PgC yr$^{-1}$ estimated with TEM during the century (Fig. 4). When considering the uncertainty
source of parameters, MIC-TEM predicted the region acts from a carbon source of 64.8 Pg C to a
carbon sink of 58.6 Pg C during the century with the mean of -3.3±20.3 Pg at the end of 21$^{st}$
century (Fig 4).






### 3.4 Model sensitivity to initial soil carbon

Under the RCP 2.6, without replacing the initial soil carbon with inventory-based estimates[1] in
model simulations, TEM estimated that the regional soil organic carbon (SOC) is 604.2 Pg C and
accumulates 12.1 Pg C during the 21[st] century. When using estimated soil carbon[1] within depths
of 30cm, 100cm and 300cm as initial pools in simulations, TEM predicted that regional SOC is
429.5, 689.3 and 1003.4 Pg C in 2000, and increases by 9.9, 16.0 and 22.8 Pg C at the end of the
21[st] century, and the regional cumulative carbon sink is 20.4, 34.0, and 48.1 Pg C, respectively
during the century. In contrast, using the same inventory-based SOC estimates, MIC-TEM
projected that the region acts from a cumulative carbon sink to a source at 0.7, 2.2, and 3.0 Pg C,
respectively. Under the RCP 8.5, both models predicted that the region acts as a carbon sink,
regardless of the magnitudes of initial soil carbon pools used, with TEM projected sink of 71.7,
120, and 155.6 Pg C and a much smaller cumulative sink of 65.4, 88.6, and 109.8 Pg C estimated
with MIC-TEM, respectively (Table 2).

### 4.  Discussion

During the last few decades, a greening accompanying warming and rising atmospheric

$CO_2$ in the northern high latitudes (>45 °N) has been documented (McGuire et al 1995, McGuire
and Hobbie 1997, Chapin and Starfield 1997, Stow et al 2004, Callaghan et al 2005, Tape et al
2006, Giorgi et al 2006). The large stocks of carbon contained in the region (Tarnocai et al 2009)
are particularly vulnerable to climate change (Schuur et al 2008, McGuire et al 2009). To date,
the degree to which the ecosystems may serve as a source or a sink of C in the future are still





uncertain (McGuire et al 2009, Wieder et al 2013). Therefore, accurate models are essential for
predicting carbon–climate feedbacks in the future (Todd-Brown et al 2013). Our regional
simulations indicate the region is currently a carbon sink, which is consistent with many previous
studies (White et al 2000, Houghton et al 2007), and this sink will grow under the RCP 8.5
scenario, but shift to a carbon source under the RCP 2.6 scenario by 2100.  MIC-TEM shows a
higher correlation between NPP and soil temperature (R=0.91) than TEM (R=0.82), suggesting
that MIC-TEM is more sensitive to environmental changes (Table S4).

Our regional estimates of carbon fluxes by MIC-TEM are within the uncertainty range

from other existing studies.  For instance, Zhuang et al 2003 estimated the region as a sink of 0.9
PgCyr[-1] in extratropical ecosystems for the 1990s, which is similar to our estimation of 0.83
PgCyr[-1] by MIC-TEM. White et al 2000 estimated that, during the 1990s, regional NEP above
50 ˚N region is 0.46 PgCyr[-1] while Qian et al 2010 estimated that NEP increased from 0 to 0.3
PgCyr[-1] for the high-latitude region above 60 ˚N during last century, and reached 0.25 PgCyr[-1]
during the 1990s. White et al 2000 predicted that, from 1850 to 2100, the region accumulated
134 PgC in terrestrial ecosystems, in comparison with our estimates of 77.6 PgC with MIC-TEM
and 89 PgC with TEM.  Our projection of a weakening sink during the second half of the 21[st]
century is consistent with previous model studies (Koven et al 2011, Schaphoff et al 2013). Our
predicted trend of NEP is very similar to the finding of White et al 2000, indicating that NEP
increases from 0.46 PgCyr[-1] in the 2000s and reaches 1.5 PgCyr[-1] in the 2070s, then decreases to
0.6 PgCyr[-1] in the 2090s.

The MIC-TEM simulated NEP generally agrees with the observations. However, model

simulations still deviate from the observed data, especially for tundra ecosystems. The deviation



may be due to the uncertainty or errors in the observed data, which do not well constrain the
model parameters. Uncertain driving data such as temperature and precipitation are also a source
of uncertainty for transient simulations.  In addition, we assumed that vegetation will not change
during the transient simulation. However, over the past few decades in the northern high latitudes,
temperature increases have led to vegetation changes (Hansen et al 2006), including latitudinal
treeline advance (Lloyd et al 2005) and increasing shrub density (Sturm et al 2001). Vegetation
can shift from one type to another because of competition for light, N and water (White et al
2000). For example, needleleaved trees tend to replace tundra gradually in response to warming.
In some areas, forests even moved several hundreds of kilometers within 100 years (Gear and
Huntley 1991). The vegetation changes will affect carbon cycling in these ecosystems. In
addition, we have not yet considered the effects of management of agriculture lands (Cole et al
1997), but Zhuang et al 2003 showed that the changes in agricultural land use in northern high
latitudes have been small.

The largest limitation to this study is that we have not explicitly considered the fire

effects. Warming in the northern high latitudes could favor fire in its frequency, intensity,
seasonality and extent (Kasischke and Turetsky 2006, Johnstone and Kasischke 2005, Soja et al
2007, Randerson,et al 2006, Bond-Lamberty et al 2007). Fire has profound effects on northern
forest ecosystems, altering the N cycle and water and energy exchanges between the atmosphere
and ecosystems. Increase in wildfires will destroy most of above-ground biomass and consume
organic soils, resulting in less carbon uptake by vegetation (Harden et al 2000), leading to a net
release of carbon in a short term. However, a suite of biophysical mechanisms of ecosystems
including post-fire increase in the surface albedo and rates of biomass accumulation may in turn,



exert a negative feedback to climate warming (Amiro et al 2006, Goetz et al 2007), further
influence the carbon exchanges between ecosystems and the atmosphere.

Moreover, carbon uptake in land ecosystems depends on new plant growth, which

connects tightly with the availability of nutrients such as mineral nitrogen. Recent studies have
shown that when soil nitrogen is in short supply, most terrestrial plants would form symbiosis
relationships with fungi; hyphae provides nitrogen to plants, in return, plants provide sugar to
fungi (Hobbie and Hobbie 2008, 2006, Schimel and Hättenschwiler 2007). This symbiosis
relationship has not been considered in our current modeling, which may lead to a large
uncertainty in our quantification of carbon and nitrogen dynamics.

Shift in microbial community structure was not considered in our model, which could

affect the temperature sensitivity of heterotrophic respiration (Stone et al 2012). Michaelis-
Menten constant ($K_m$) could also adapt to climate warming, and it may increase more
significantly with increasing temperature in cold-adapted enzymes than in warm-adapted
enzymes (German et al 2012, Somero et al 2004, Dong and Somero 2009). Carbon use efficiency
(CUE) is also a controversial parameter in our model. Empirical studies in soils suggest that
microbial CUE declines by at least 0.009 $°C^{-1}$ (Steinweg et al 2008), while other studies find that
CUE is invariant with temperature (López-Urrutia and Morán 2007). Another key microbial trait
lacking in our modeling is microbial dormancy (He et al 2015). Dormancy is a common, bet-
hedging strategy used by microorganisms when environmental conditions limit their growth and
reproduction (Lennon and Jones 2011). Microorganisms in dormancy are not able to drive
biogeochemical processes such as soil $CO_2$ production, and therefore, only active
microorganisms should be involved in utilizing substrates in soils (Blagodatskaya and Kuzyakov



2013). Many studies have indicated that soil respiration responses to environmental conditions
are more closely associated with the active portion of microbial biomass than total microbial
biomass (Hagerty et al 2014, Schimel and Schaeffer 2012, Steinweg et al 2013). Thus, the
ignorance of microbial dormancy could fail to distinguish microbes with different physiological
states, introducing uncertainties to our carbon estimation.
**5.  Conclusions**
This study used a more detailed microbial biogeochemistry model to investigate the carbon
dynamics in the region for the past and this century. Regional simulations using MIC-TEM
indicated that, over the 20th century, the region is a sink of 77.6 Pg. This sink could reach to 79.5
Pg under the RCP 8.5 scenario or shift to a carbon source of 1.6 Pg under the RCP 2.6 scenario
during 21st century. On the other hand, traditional TEM overestimated the carbon sink under the
RCP 8.5 scenario with magnitude of 19.2 Pg than MIC-TEM, and predicted this region acting as
carbon sink with magnitude of 27.6 Pg under the RCP 2.6 scenario during 21st century. Using
recent soil carbon stock data as initial soil carbon in model simulations, the region was estimated
to shift from a carbon sink to a source, with total carbon release at 0.7- 3 Pg by 2100 depending
on initial soil carbon pools at different soil depths under the RCP 2.6 scenario. In contrast, the
region acts as a carbon sink at 55.4 - 99.8 Pg C in the 21st century under RCP 8.5 scenario.
Without considering more detailed microbial processes, models estimated that the region acts as
a carbon sink under both scenarios.  Under the RCP 2.6 scenario, the cumulative sink ranges
from 9.9 to 22.8 Pg C.  Under the RCP 8.5 scenario, the cumulative sink is even larger at 71.7 -
155.6 Pg C.  This study indicated that more detailed microbial physiology-based
biogeochemistry models estimate carbon dynamics very differently from using a relatively



simple microbial decomposition-based model.  The comparison with satellite products or other
estimates for the 20th century suggests that the more detailed microbial decomposition shall be
considered to adequately quantify C dynamics in northern high latitudes.

**Acknowledgments**
This research was supported by a NSF project (IIS-1027955), a DOE project (DE-SC0008092),
and a NASA LCLUC project (NNX09AI26G) to Q. Z. We acknowledge the Rosen High
Performance Computing Center at Purdue for computing support. We thank the National Snow
and Ice Data center for providing Global Monthly EASE-Grid Snow Water Equivalent data,
National Oceanic and Atmospheric Administration for North American Regional Reanalysis
(NARR), and Hugelius and his group by making available pan-Arctic permafrost soil C maps.
We also acknowledge the World Climate Research Programme's Working Group on Coupled
Modeling Intercomparison Project CMIP5, and we thank the climate modeling groups for
producing and making available their model output. The data presented in this paper can be
accessed through our research website (http://www.eaps.purdue.edu/ebdl/)










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

**Author contributions.** Q.Z. designed the study. J.Z. conducted model development, simulation
and analysis. J.Z. and Q. Z. wrote the paper.
**Competing financial interests.** The submission has no competing financial interests.
**Materials & Correspondence.** Correspondence and material requests should be addressed to
qzhuang@purdue.edu.




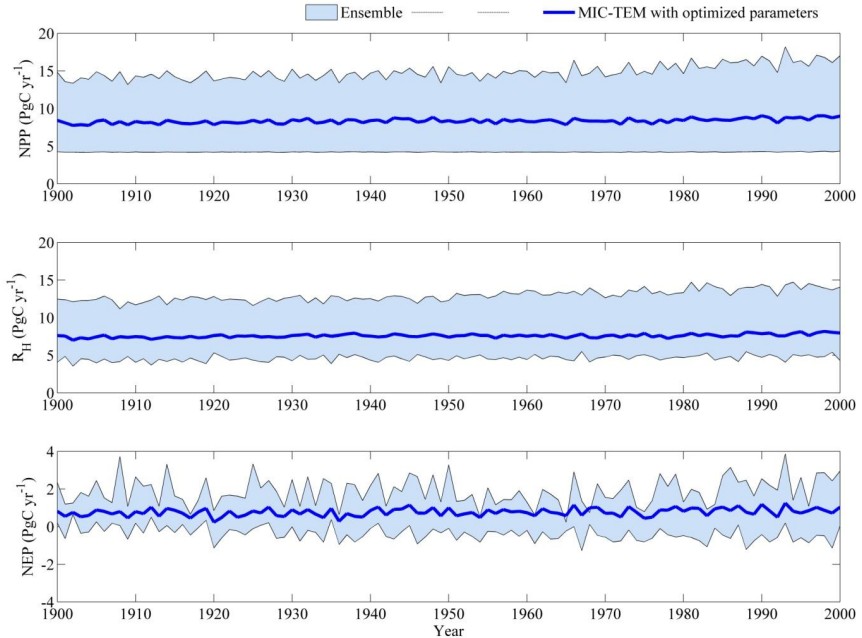

Figure 1. Simulated annual net primary production (NPP, top panel), heterotrophic respiration
($R_H$, center panel) and net ecosystem production (NEP, bottom panel) by MIC-TEM with
ensemble of parameters.




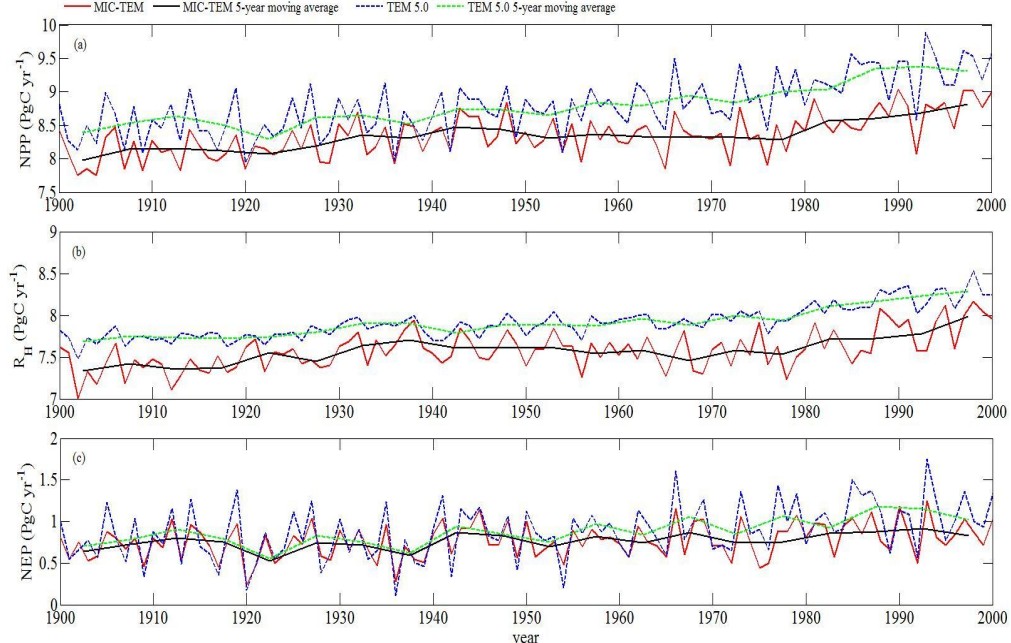

Figure 2. Simulated annual net primary production (NPP, top panel), heterotrophic respiration ($R_H$, center panel) and net ecosystem production (NEP, bottom panel) by MIC-TEM and TEM, respectively.




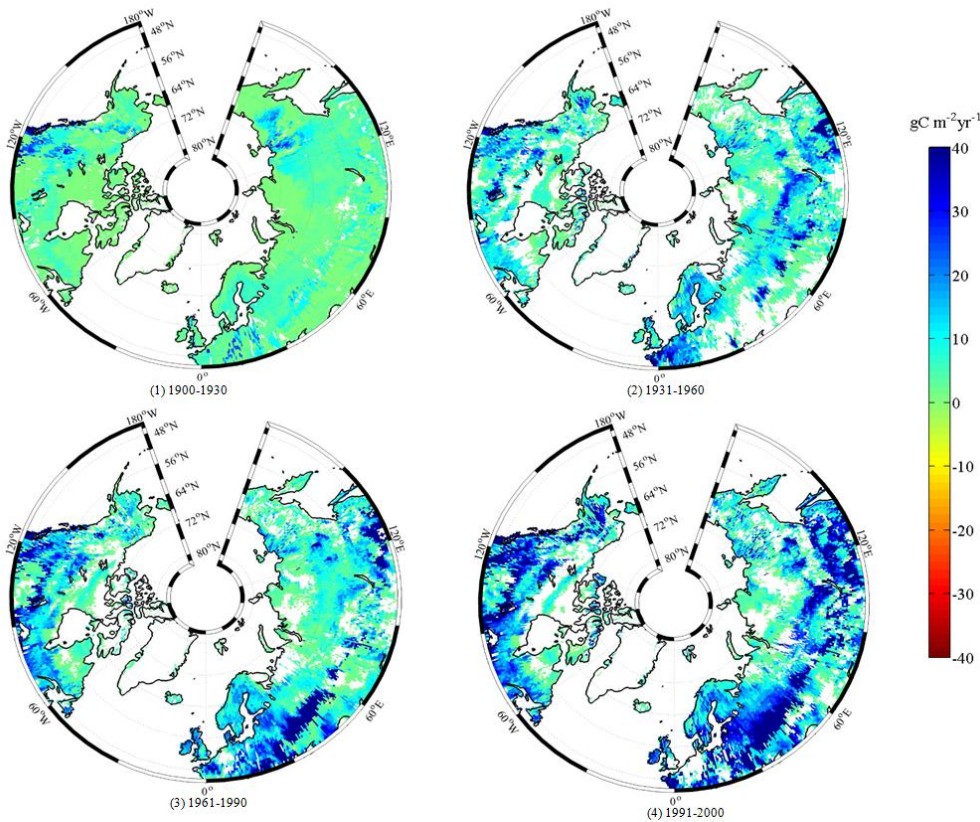

Figure 3. Spatial distribution of NEP simulated by MIC-TEM for the periods: (1) 1900-1930,
(2) 1931-1960, (3) 1961-1990, and (4) 1991-2000. Positive values of NEP represent sinks of
$CO_2$ into terrestrial ecosystems, while negative values represent sources of $CO_2$ to the
atmosphere.




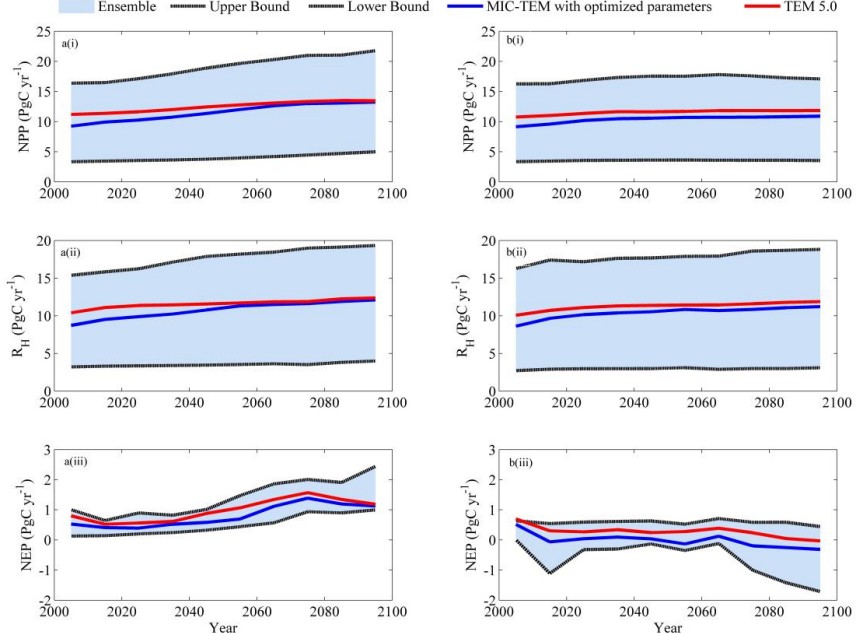

Figure 4. Predicted changes in carbon fluxes: (i) NPP, (ii) $R_H$, and (iii) NEP for all land areas north of 45 °N in response to transient climate change under (a) RCP 8.5 scenario and (b) RCP 2.6 scenario with MIC-TEM and TEM 5.0, respectively. The decadal running mean is applied. The grey area represents the upper and lower bounds of simulations.





**Table 1. Partitioning of average annual net ecosystem production (as Pg C per year) for six vegetation types during the 20th century**

|  | MIC-TEM (PgC yr⁻¹) | TEM 5.0 (PgC y⁻¹) |
|---|---|---|
| Alpine tundra | 0.03 | 0.04 |
| Boreal forest | 0.39 | 0.45 |
| Conifer forest | 0.09 | 0.09 |
| Deciduous forest | 0.16 | 0.18 |
| Grassland | 0.06 | 0.07 |
| Wet tundra | 0.05 | 0.06 |
| Total | 0.78 | 0.89 |



**Table 2. Increasing of SOC, vegetation carbon (VGC), soil organic nitrogen (SON), vegetation nitrogen (VGN) from 1900 to 2000, and total carbon storage during the 21$^{st}$ century predicted by two models with observed soil carbon data of three different depths under (a) RCP 2.6 and (b) RCP 8.5.**

(a)

| Model | Units: Pg | Without (control) | 30cm | 100cm | 300cm |
|---|---|---|---|---|---|
| TEM 5.0 | SOC/SON in 2000 | 604.2/27.0 | 429.5/19.0 | 689.3/31.6 | 1003.4/46.2 |
| | Increase of SOC during the 21$^{st}$ century | 12.1 | 9.9 | 16.0 | 22.8 |
| | VGC/VGN in 2000 | 318.3/1.48 | 238.4/1.05 | 394.2/1.80 | 556.7/2.53 |
| | Increase of VGC during the 21$^{st}$ century | 15.5 | 10.5 | 18.0 | 25.3 |
| | Increase of total carbon storage during the 21$^{st}$ century | 27.6 | 20.4 | 34.0 | 48.1 |
| MIC-TEM | SOC/SON in 2000 | 591.5/26.8 | 420.3/18.6 | 686.0/31.2 | 990.7/45.3 |
| | Increase of SOC during the 21$^{st}$ century | -2.0 | -1.2 | -2.4 | -2.9 |
| | VGC/VGN in 2000 | 309.7/1.42 | 230.1/1.02 | 374.4/1.71 | 548.6/2.45 |
| | Increase of VGC during the 21$^{st}$ century | 0.4 | 0.5 | 0.2 | -0.1 |
| | Increase of total carbon storage during the 21$^{st}$ century | -1.6 | -0.7 | -2.2 | -3.0 |

(b)

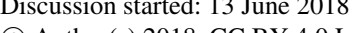



| Model | Units: Pg | Without (control) | 30cm | 100cm | 300cm |
|---|---|---|---|---|---|
| TEM 5.0 | SOC/SON in 2000 | 610.2 /27.9 | 431.9/19.1 | 693.8/31.8 | 1007.1/46.4 |
| | Increase of SOC during the 21$^{st}$ century | 44.2 | 33.0 | 56.5 | 74.6 |
| | VGC/VGN in 2000 | 324.9/1.50 | 242.1/1.07 | 399.6/1.83 | 570.2/2.57 |
| | Increase of VGC during the 21$^{st}$ century | 54.5 | 38.7 | 63.5 | 81.0 |
| | Increase of total carbon storage during the 21$^{st}$ century | 98.7 | 71.7 | 120.0 | 155.6 |
| MIC-TEM | SOC/SON in 2000 | 596.0/27.1 | 424.6/18.8 | 689.1/31.5 | 995.5/46.1 |
| | Increase of SOC during the 21$^{st}$ century | 33.3 | 27.4 | 36.9 | 42.9 |
| | VGC/VGN in 2000 | 316.0/1.44 | 233.5/1.02 | 380.0/1.72 | 568.3/2.56 |
| | Increase of VGC during the 21$^{st}$ century | 46.2 | 37.0 | 51.7 | 56.9 |
| | Increase of total carbon storage during the 21$^{st}$ century | 79.5 | 65.4 | 88.6 | 109.8 |