# Peer review of "1. Introduction"

_Biogeosciences, 2018_

## Referee Comment (RC1) · Anonymous Referee #1 · 6 Jul 2018

Summary. Authors implement simple yet effective model of microbial biomass dynamics, that improves NPP/NEE seasonal cycle simulation by modified TEM (MIC-TEM) model for several observation sites in Arctic. They also apply the modified mode to simulating the trends of the soil carbon storage under climate change. The study is valuable and manuscript is reasonably well written, so it can be accepted after implementing suggested revisions.

General comments.

1. It appears that the transient simulations for 20th and 21st century runs are starting from non-equilibrium state, initialized from observations. That introduces artificial dis-

turbance likely to affect conclusions on ecosystem carbon storage trends. Additional tests with well equilibrated initial state are needed to clarify the potential problem.

2. Model description contains several deficiencies and omissions, that need to be corrected (see detailed comments).

3. Model parameters are not presented, a table of the model parameters should be added

4. References in a manuscript and the supplement should be formatted according to Biogeoscience journal format

Detailed comments.

Line 140 (L140) Abbreviation DOC is used, so it should be introduced here rather than at Line 155

L150 "microbial biomass death (DEATH) and enzyme production (EPROD) are modeled as constant fraction of microbial biomass". According to Eq. 6, DEATH appears as a process rate, so it cannot be a fraction of MIC, it can be proportional to MIC. To avoid confusion, authors need to rewrite the Eq. 6 in terms of monthly increments (delta MIC), not as process rates (dMIC/dt).

L152 Formally, if Eq. 6 is right, in Eq. 7 DEATH should appear as a multiple of MIC and a process rate constant, the rate constant (units: sec-1) is missing, the rdeath is a ratio, assumed non-dimensional. Same problem with Eq. 8. Authors should explain what is in fact meaning of DEATH and EPROD, is it a process rate (as appears in Eq. 6) or (monthly) increment due to the conversion from one (organic matter) pool to another?

L157 "MICtoSOC is carbon input" – suggest to write "MICtoSOC is carbon input ratio"

L170 KmðİŠăðİŚŹðİŚIJðİŚİðİŚŠ not explained.

L189 The source of MODIS NPP (version, MODIS product name and parameter) are not mentioned.

L225-236 Using non-equilibrium initial SOC taken from observations cannot be recommended for transient simulations, even for a model like TEM, that doesn't have very slow soil carbon pools. Accordingly, additional tests should be made with equilibrated initial SOC set by long enough spinup run (200-300 years) to the equilibrium.

L436-L720 References should be formatted according to Biogeoscience format.

Supplementary material:

Formatting of references should be fixed to same style as paper (also check initials vs full name)

---

## Referee Comment (RC2) · Anonymous Referee #2 · 25 Jul 2018

Summary

The authors present a microbe-based biogeochemistry model (MIC-TEM) based on an extant Terrestrial Ecosystem Model (TEM). The MIC-TEM heterotrophic respiration is calculated taking into account the influence of the dynamics of microbial biomass and enzyme kinetics. The verified MIC-TEM was used to quantify the regional carbon dynamics in northern high latitudes (north 45 °N) during the 20th and 21st centuries. It is very important that as a rule the models is account climate change but not take account many other parameters which may significantly change the overall picture in conducting global or regional assessments - namely, changes in the vegetation cover

and microbial communities as a result of climate change, land use change, fires, etc.. The study is valuable and manuscript is well written enough, so it can be accepted after minor revisions.

General comments. L 76. (here and hereinafter in the text) - "Most models treated soil decomposition as a first-order decay process, i.e., CO2 respiration is directly proportional to soil organic carbon." The region chosen for modeling is very large. There are ecosystems with very different reserves of SOC on this territory. In reality, there can be no direct dependence of respiration CO2 from SOC. The main and most active processes associated with the transformation of organic carbon and emissions occur mainly in the upper horizons of soils. The authors try to take into account the carbon stocks at different depths of 30, 100 and 300 cm, and according to the model - the more carbon stocks the more it accumulates. However, northern high latitude ecosystems are often represented by wetlands with large organic carbon stocks in the form of peat deposits. While most of them have low productivity, in contrast to boreal forests, where the stock of soil carbon is much lower. Is this taken into account when modeling?

Detailed comments. In the abstract, there is no mention of the improved model (only TEM), therefore it is not clear on which model the values of sink or source of carbon were obtained. Some of model parameters are not presented, a table of the model parameters should be added for example how litterfall is calculated. It is not clear what territory is taken for modeling - in the name and abstract of article are talking about the Arctic ecosystems, in the Fig. 3 and Fig.S5 represent the territory of the exciting 45 N, in the text 45 °N or 60 °N - which territory was being investigated? In fact, a period of 200 years (20th century and 21st century) is simulated, which SOC value was taken as the initial value. A value characteristic of 2000 yr or what? When modeling the 20th century, which parameters of the model were taken as input? The source of MODIS NPP (version, MODIS product name and parameter) are not mentioned. It is also not clear how values of NPP were obtained by model TEM and MIC-TEM.

---

## Author Comment (AC1) · 9 Aug 2018

We thank the Associate Editor and two referees for their providing constructive comments to this manuscript. Below we detail how we have revised the manuscript following their suggestions.

1. It appears that the transient simulations for 20th and 21st century runs are starting from non-equilibrium state, initialized from observations. That introduces artificial disturbance likely to affect conclusions on ecosystem carbon storage trends. Additional tests with well equilibrated initial state are needed to clarify the potential problem.

Response: Thanks for the comments. We would like to clarify that, we actually initialized the model with the observation-based SOC for transient simulations during the 20th and 21st centuries to compare with the simulations that initialized with equilibrated state. These simulations were presented in Table 3.

2. Model description contains several deficiencies and omissions that need to be corrected (see detailed comments).

Response: Thanks. We have provided more detailed model description. See the manuscript lines 140-202.

3. Model parameters are not presented, a table of the model parameters should be added.

Response: Thanks for the suggestion. In this revision, we moved the model parameter table from supplementary materials to main text, as Table 1.

4. References in a manuscript and the supplement should be formatted according to Biogeoscience journal format.

Response: Thanks. In this revision, we changed the format according to Biogeoscience journal format.

Detailed comments: Line 140 (L140) Abbreviation DOC is used, so it should be introduced here rather than at Line 155.

Response: Yes, we have specified the abbreviation when its first appearance.

L150 "microbial biomass death (DEATH) and enzyme production (EPROD) are modeled as constant fraction of microbial biomass". According to Eq. 6, DEATH appears as a process rate, so it cannot be a fraction of MIC, it can be proportional to MIC. To avoid confusion, authors need to rewrite the Eq. 6 in terms of monthly increments (delta MIC), not as process rates (dMIC/dt).

Response: Thanks. In this revision, we stated that both microbial biomass death
(DEATH) and enzyme production (EPROD) are modeled as proportional to microbial biomass with constant rates. rdeath and rEnzProd are rate constants. Thus, the formula doesn't need to be changed.

L152 Formally, if Eq. 6 is right, in Eq. 7 DEATH should appear as a multiple of MIC and a process rate constant, the rate constant (units: sec-1) is missing, the rdeath is a ratio, assumed non-dimensional. Same problem with Eq. 8. Authors shou Id explain what is in fact meaning of DEATH and EPROD, is it a process rate (as appears in Eq. 6) or (monthly) increment due to the conversion from one (organic matter) pool to another?

Response: Similar to above, we have stated that both microbial biomass death (DEATH) and enzyme production (EPROD) are modeled as proportional to microbial biomass with constant rates rdeath and rEnzProd.

L157 "MICtoSOC is carbon input" - suggest to write "MICtoSOC is carbon input ratio"

Response: Yes. We have changed it according to your suggestion.

L170 Km not explained.

Response: In this revision, we explained how Km is calculated.

L189 The source of MODIS NPP (version, MODIS product name and parameter) are not mentioned.

Response: The MODIS NPP data was derived by the MOD17 MODIS project. The product name is Net Primary Production Yearly L4 Global 1 km. The critical parameter used in MOD17 algorithm is conversion efficiency parameter ÆŘ. More information about the MODIS NPP product could be found on https://neo.sci.gsfc.nasa.gov/view.php?datasetId=MOD17A2\_M\_PSN. In this revision, we added this information into main text in lines 193 – 202.

L225-236 Using non-equilibrium initial SOC taken from observations cannot be recommended for transient simulations, even for a model like TEM, that doesn't have very

BGD
slow soil carbon pools. Accordingly, additional tests should be made with equilibrated initial SOC set by long enough spinup run (200-300 years) to the equilibrium.

Response: See our above response about comparison between these two types of simulations.

Please also note the supplement to this comment: https://www.biogeosciences-discuss.net/bg-2018-241/bg-2018-241-AC1supplement.pdf

---

## Author Comment (AC2) · 9 Aug 2018

We thank the Associate Editor and two referees for their providing constructive comments to this manuscript. Below we detail how we have revised the manuscript following their suggestions.

General comments. L 76. (here and hereinafter in the text) - "Most models treated soil decomposition as a first-order decay process, i.e., $CO_2$ respiration is directly proportional to soil organic carbon." The region chosen for modeling is very large. There are ecosystems with very different reserves of SOC on this territory. In reality, there can be no direct dependence of respiration $CO_2$ from SOC. The main and most active processes associated with the transformation of organic carbon and emissions occur mainly in the upper horizons of soils. The authors try to take into account the carbon stocks at different depths of 30, 100 and 300 cm, and according to the model - the more carbon stocks the more it accumulates. However, northern high latitude ecosystems are often represented by wetlands with large organic carbon stocks in the form of peat deposits. While most of them have low productivity, in contrast to boreal forests, where the stock of soil carbon is much lower. Is this taken into account when modeling?

Response: Thanks. The pixels in our model were not split into uplands and peatlands ecosystems. All the carbon pools represent the total amount of carbon for each pixel on a per unit area basis. Similarly, the inventory or observation-based estimates of SOC from Tarnocai et al. (2009) also covers both uplands and lowlands / wetlands across the landscape without explicitly differentiating these land types.

In the abstract, there is no mention of the improved model (only TEM), therefore it is not clear on which model the values of sink or source of carbon were obtained.

Response: Thanks. In this revision, we clarified that the results are simulated from new model, which is MIC-TEM.

Some of model parameters are not presented, a table of the model parameters should be added for example how litterfall is calculated.

Response: Thanks. In this revision, we moved the table from supplementary materials to main text, as Table 1.

It is not clear what territory is taken for modeling - in the name and abstract of article are talking about the Arctic ecosystems, in the Fig. 3 and Fig.S5 represent the territory of the exciting 45 N, in the text 45 âŮẹN or 60 âŮẹN - which territory was being investigated?

Response: Thanks. In this revision, we stated that our study region is north 45°N above.

In fact, a period of 200 years (20th century and 21st century) is simulated, which SOC value was taken as the initial value. A value characteristic of 2000 yr or what? When modeling the 20th century, which parameters of the model were taken as input?

Response: The transient simulations for the 20th century were initialized with equilibrated state after spin-up. The initial state variables include vegetation carbon, soil carbon, vegetation nitrogen, soil organic carbon, and the total soil inorganic nitrogen that are obtained after equilibrium, which typically takes several hundred years in TEM (See Qu et al., 2018).

Qu, Y., Maksyutov, S., and Zhuang, Q. Technical Note: An efficient method for accelerating the spin-up process for process-based biogeochemistry models, Biogeosciences, 15, 3967–3973, 2018 https://doi.org/10.5194/bg-15-3967-2018

The source of MODIS NPP (version, MODIS product name and parameter) are not mentioned. It is also not clear how values of NPP were obtained by model TEM and MIC-TEM.

Response: In this revision, we added the following to main text "The MODIS NPP data was developed by the MOD17 MODIS project. The product name is Net Primary Production Yearly L4 Global 1 km. The critical parameter used in MOD17 algorithm is conversion efficiency parameter ÆŘ. More information about the MODIS NPP product could be found on https://neo.sci.gsfc.nasa.gov/view.php?datasetId=MOD17A2_M_PSN".

Please also note the supplement to this comment:
https://www.biogeosciences-discuss.net/bg-2018-241/bg-2018-241-AC2-supplement.pdf

**Supplement:**

[revised manuscript text omitted]

:

$$\text{DEATH} = r_{\text{death}} * \text{MIC} \qquad\qquad (7)$$

$$\text{EPROD} = r_{\text{EnzProd}} * \text{MIC} \qquad\qquad (8)$$

Where $r_{\text{death}}$ and $r_{\text{EnzProd}}$ are the ratio of microbial death and enzyme production, respectively.

DOC is part of soil organic carbon:

$$\frac{\text{dDOC}}{\text{dt}} = \text{DEATH} * (1 - \text{MICtoSOC}) + \text{DECAY} + \text{ELOSS} - \text{ASSIM} \qquad (9)$$

where MICtoSOC is carbon input ratio as dead microbial biomass to SOC, representing the fraction of microbial death that flows into SOC, and is set as a constant value according to Allison et al. (2010). SOC dynamics are modeled:

$$\frac{\text{dSOC}}{\text{dt}} = \text{Litterfall} + \text{DEATH} * \text{MICtoSOC} - \text{DECAY} \qquad\qquad (10)$$

Where Litterfall is estimated as a function of vegetation carbon (Zhuang et al., 2010). The enzymatic decay of SOC is calculated as:

$$\text{DECAY} = V_{\text{max}} * \text{ENZ} * \frac{\text{SOC}}{Km + \text{SOC}} \qquad\qquad (11)$$

Where $V_{\text{max}}$ is the maximum velocity of the reaction and calculated using the Arrhenius equation:

$$V_{max} = \text{Vmax}_0 * e^{\frac{Ea}{R*(\text{temp}+273)}} \qquad\qquad (12)$$

The parameters Km and carbon use efficiency (CUE) are temperature sensitive, and calculated as a linear function of temperature between 0 and 50°C:

$$Km = \text{Km}_{slope} * \text{temp} + \text{Km}_0 \qquad\qquad (13)$$

[revised manuscript text omitted]

---

## Author Comment (AC3) · 9 Aug 2018

For tracking changes, please see the attached revised manuscript.

Please also note the supplement to this comment:
https://www.biogeosciences-discuss.net/bg-2018-241/bg-2018-241-AC3-supplement.pdf

―――――――――――――――――――――

---

## Author Response (AR2)

We thank the Associate Editor for providing constructive comments to this manuscript. Below we detail how we have revised the manuscript following your suggestions.

1. L68 (and abstract) Suggest to modify wording "inadequate [representation]" because it is applied in general scope, risking to blame all available models, without mentioning that the models of higher level of complexity (eg DNDC) do exist and are being applied with success, although in more local, regional or site level scale.

*Response: Thanks for the comment. We have revised the first two sentences to "Various levels of representation of biogeochemical processes in current biogeochemistry models contribute to a large uncertainty in carbon budget quantification. Here, we present an uncertainty analysis with a process-based biogeochemistry model, the Terrestrial Ecosystem Model (TEM), that was incorporated with detailed microbial mechanisms"*

2. L112 Q10 is wrongly defined as "soil temperature (Q10)"

*Response: Thanks for the comment. We have modified the definition of $Q_{10}$ as "temperature sensitivity of heterotrophic soil respiration".*

3. L125 soil temperature DT definition is vague (could be (T-273)?)

*Response: Thanks for the comment. We have modified the term "DT" as "temp", which is consistent with the content below. The term "temp" represents soil temperature at top 20 cm, and the units are ℃.*

4. L144 Another temperature is defined as temp, is it different from DT?

*Response: Thanks for the comment. The term "temp" represents soil temperature at top 20 cm, and the units are ℃. It is the same as previous "DT". And we have modified the term "DT" to "temp" to make them consistent.*

5. L151 Better to call r 'rate constant', not constant rate.

*Response: Thanks for the comment. We have changed r to "rate constant".*

6. L154 Definition "rdeath and rEnzProd are the ratio of microbial death and enzyme production" contradicts L151 where both are defined as "constant rates"

*Response: Thanks for the comment. We have changed "the ratio" to "rate constants".*

7. L175 renzloss is defined as the ratio of enzyme loss, while it looks more like rate
constant

*Response: Thanks for the comment. We have changed r to "rate constant".*

8. L199 need to explain factors in eq 18.

*Response: Thanks for the comment. We have added the explanation for factors and
functions "Where Cmax is the maximum rate of carbon assimilation, PAR is
photosynthetically active radiation, and f(phenology) represents the effects of leaf area
(Raich et al., 1991). The function f(foliage) represents the ratio of canopy leaf biomass
relative to maximum leaf biomass (Zhuang et al., 2002). T is monthly air temperature,
and f(CO2) represents the effects of elevated atmospheric CO2 (McGuire et al., 1997;
Pan et al.,1998). The function f(NA) models the limiting effects of plant nitrogen status on
GPP (McGuire et al., 1992; Pan et al., 1998). The function f (FT) represents the effects
of freeze-thaw (Zhuang et al., 2003).".*

9. L436 Suggest adding initials to persons name.

*Response:* We didn't use his data and we deleted that sentence.

10. English should be checked carefully, especially in sections 2.2, 2.3, 3.3

*Response:  We carefully checked English for the manuscript, especially for these sections
in this revision.*

[revised manuscript text omitted]